# Metabolomics Revealed the Tolerance and Growth Dynamics of Arbuscular Mycorrhizal Fungi (AMF) to Soil Salinity in Licorice

**DOI:** 10.3390/plants13182652

**Published:** 2024-09-22

**Authors:** Li Fan, Chen Zhang, Jiafeng Li, Zhongtao Zhao, Yan Liu

**Affiliations:** 1Horticultural and Crop Protection College, Inner Mongolia Agricultural University, Hohhot 010018, China; fanli@imau.edu.cn (L.F.); zhch1026@163.com (C.Z.); jia.feng1999@163.com (J.L.); 2South China Botanical Garden, Chinese Academy of Sciences, Guangzhou 510520, China; zhzht621@scbg.ac.cn

**Keywords:** saline–alkali soil, licorice, root, AMF, metabolomics

## Abstract

Several studies have been devoted to seeking some beneficial plant-related microorganisms for a long time, and on this basis, it has been found that arbuscular mycorrhizal fungi (AMF) have a considerable positive impact on plant health as a biological fungal agent. In this study, we focused on the effects of different AMF on the growth dynamics and root configuration of licorice under saline and alkali conditions. The metabolites of licorice under different AMF were assessed using liquid chromatography–tandem mass spectrometry (LC-MS/MS). *Funneliformis mosseae* (Fm) and *Rhizophagus intraradices* (Ri) were added as different AMF treatments, while the sterilized saline–alkali soil was treated as a control. Samples were taken in the R1 period (15 d after AMF treatment) and the R2 period (45 d after AMF treatment). The results showed that the application of AMF significantly increased the root growth of licorice and significantly increased the biomass of both shoot and root. A total of 978 metabolites were detected and divided into 12 groups including lipids, which accounted for 15.44%; organic acids and their derivatives, at 5.83%; benzene compounds and organic heterocyclic compounds, at 5.42%; organic oxides, at 3.78%; and ketones, accounting for 3.17%. Compared with the control, there were significant changes in the differential metabolites with treatment inoculated with AMF; the metabolic pathways and biosynthesis of secondary metabolites were the main differential metabolite enrichment pathways in the R1 period, and those in the R2 period were microbial metabolism in diverse environments and the degradation of aromatic compounds. In conclusion, the use of AMF as biofertilizer can effectively improve the growth of licorice, especially in terms of the root development and metabolites, in saline–alkali soil conditions.

## 1. Introduction

Soil salinization is a phenomenon of soil quality deterioration caused by the accumulation of dissolved salts in the surface layer of the soil, leading to an increase in soil pH and a consequent decrease in soil structure, fertility, and microbial activity. Soil salinization is one of the important problems that restrict the development of agricultural production and threaten ecological environment security [1]. According to the United Nations Educational Scientific and Cultural Organization (UNESCO) and the World Food Organization (FAO), soil salinization is still on the rise worldwide. More than 100 countries have been affected by soil salinization, with a total global saline land area of about 954 million hm^2^, mainly in Africa, the western part of the Americas, and Eurasia [2]. China’s saline–alkali land has a large area and wide distribution, and the causes of formation are complex and diverse. It ranks third in the world in the ranking of saline–alkali land countries, with a total saline–alkali land area of 99.13 million hm^2^ [3]. In order to promote the treatment and transformation of saline–alkali land in China and the efficient use of land, we have broken through the mechanism of the efficient reduction of soil salt and alkali barriers and increased carbon for fertilizer cultivation. With increasing saline levels, lower biomass, reduced plant height, and less enzyme activity were reported [4]. Biological management is the most sustainable and friendly approach to saline–alkali land management. Through the application of technological achievements, saline–alkali land management, utilization, promotion, and eco-friendly coordination can be achieved [5].

Arbuscular mycorrhizal fungi (AMF) are widely distributed and can form symbiotic systems with most land plants [6]. These fungi are able to extend the root uptake range of plants, helping them to access water and minerals more efficiently. The mycelial network delivers nutrients from the soil to the plant, while the plant provides the fungi with a source of organic carbon [7,8]. AMF can significantly improve plant resistance to diseases, drought stress, salt stress, etc., through a series of non-nutritional pathways and enhance the growth potential of the plant [9,10]. At the community level, mycorrhizal fungi alter plant diversity–productivity relationships by acting on plant–soil feedback [11], influencing nutrient uptake efficiency, and influencing plant community productivity [12,13,14]. Enhanced nutrient uptake by plants also promotes the cycling of nutrients in the soil and the improvement in soil structure, which plays a vital role in maintaining the health and productivity of ecosystems.

Licorice (*Glycyrrhiza uralensis* F.) is a perennial persistent herb in the genus Glycyrrhiza of the family Leguminosae, and is also known as Guo Lao and Sweetgrass. It is one of the famous traditional bulk Chinese medicinal herbs in China [15], and its root and rhizome extracts contain glycyrrhizic acid and flavonoids, which have anti-cancer [16] and anti-viral effects [17]. Licorice is distributed mainly in north China, especially in the provinces of Xinjiang, Gansu, Inner Mongolia, and Ningxia [18]. Because of its characteristics of drought resistance and salinity resistance, the natural habitat of licorice is often mildly, moderately, or heavily salinized, and it is a typical saline plant [19,20]. Due to its economic value, licorice has been listed as a national Class II endangered key protected plant [21]. In recent years, AMF have been increasingly used as a biofertilizer in agriculture. Parihar et al. reported that the action of various AMF complexes altered the growth, yield performance, and metabolic changes of pea crop and mitigated the negative effects of pea under salt stress conditions [22]. But fewer studies have been conducted on licorice under saline and alkaline stress inoculated with AMF. Therefore, this study focused on the growth dynamics and changes in metabolites of licorice at the seedling stage under saline and alkaline stresses with different AMF inoculation amounts. Therefore, the research provides a new strategy for the application of microbial fertilizers in saline–alkali land with the sustainable utilization of licorice resources.

## 2. Results

### 2.1. Licorice Growth Dynamics and Root AMF Inoculation Index Changes

AMF inoculation promoted biomass production in aboveground parts and roots, which was related to the species of AMF (Figure 1a–d). After inoculation with *Funneliformis mosseae* (Fm) and *Rhizophagus intraradices* (Ri), the shoot biomass of licorice was 95.52% and 57.27% higher than CK in the R2 period, respectively, and the root biomass of licorice was 75.80% and 71.64% higher than CK in the R2 period. The roots of non-mycorrhizal plants exhibited no mycorrhizal colonization. The root mycorrhiza colonization rate was 40.47~46.12% in the AMF treatment group. In the R2 period, the Fm treatment increased the root mycorrhiza colonization rate by 13.96% compared with the Ri treatment (Figure 1e).

### 2.2. Root System Architecture Analysis

Phenotypic adaptation is the response of roots to the local nutrient supply in soil and to other environmental conditions. It is the key to determining the distribution of root systems in soil and the efficiency of nutrient absorption by plants. The root structural characteristics of licorice were affected by AMF species (Figure 2a,b). With the change of time, the structural characteristics of the root system changed obviously. Under the influence of different bactericides, Fm and Ri both showed certain advantages compared with CK. In the R2 period, the total root length, root volume, and number of root branches increased by 105.51% and 39.95%, 116.00% and 52.00%, and 127.22% and 46.33%, respectively. Fm was more efficient in enhancing the morphological characteristics of licorice compared with Ri for each trait (Table 1). 

### 2.3. Metabolite Detection

Figure 3 shows the results of the analysis to quantify the changes in the metabolite content and metabolic pathway of glycyrrhiza licorice under different inoculation conditions. Based on the UPLC-MS/MS detection platform and database, a total of 978 metabolites were detected (Figure 3). The classification at the material level was divided into 12 groups. Lipids accounted for 15.44% (151), organic acids and their derivatives accounted for 5.83% (57), benzene compounds and organic heterocyclic compounds accounted for 5.42% (53), organic oxides accounted for 3.78% (37), ketones accounted for 3.17% (31), organic nitrides accounted for 1.43% (14), organic sulfide accounted for 0.31% (3), nucleotides and their derivatives and alkaloids and their derivatives accounted for 0.20% (2), and lignin-related compounds accounted for 0.1% (1).

### 2.4. Changes in Root Differential Metabolites

In the R1 period, in the CK_vs._Fm treatment, a total of 82 differential metabolites were identified, among which 68 metabolites were down-regulated and 14 metabolites were up-regulated (Figure 4a). In the R1 period, in the CK_vs._Ri treatment, a total of 73 differential metabolites were identified, among which 20 metabolites were up-regulated and 53 metabolites were down-regulated (Figure 4b). In the R2 period, in the CK_vs._Fm treatment, a total of 143 differential metabolites were identified, among which 24 metabolites were up-regulated and 119 metabolites were down-regulated (Figure 4c). In the R2 period, in the CK_vs._Fm treatment, a total of 104 differential metabolites were identified, among which 14 metabolites were up-regulated and 90 metabolites were down-regulated (Figure 4d). Differential metabolites often have biological similarity or complementarity in terms of outcomes and functions, and are also positively/negatively regulated by the same metabolic pathways. Hierarchical cluster analysis helps to group metabolites with the same characteristics and to discover the characteristics of metabolite variation between treatments. Figure 5 clearly shows the differences in the metabolites and the hierarchical clustering results of these related differences between treatments.

The top 20 differential metabolites are shown in Figure 6. The up-regulated metabolites were selected mainly through a log_2_Fold-change > 1, while the down-regulated metabolites were selected mainly through a log_2_Fold-change < −1. In the R1 period, in the CK_vs._Fm treatment, the up-regulated differential metabolites were arranged according to the following trend in descending order: Xanthine > Oxypurinol. The down-regulated differential metabolites were arranged according to the following trend in descending order: Norlichexanthone > 7-Chloro-5-phenyl-2,3,4,5-tetrahydro-1H-1,4-benzodiazepin-2-one > (-)-O-Acetyl-D-mandelic acid > Coniferyl acetate > Mitragynine (Figure 6a). In the R1 period, in the CK_vs._Ri treatment, the up-regulated differential metabolites were arranged according to the following trend in descending order: Xanthine > Oxypurinol > 2-[(7-Chloro-4-nitro-2,1,3-benzoxadiazol-5-yl)amino]ethanol. The down-regulated differential metabolites were arranged according to the following trend in descending order: 7-Chloro-5-phenyl-2,3,4,5-tetrahydro-1H-1,4-benzodiazepin-2-one > Mitragynine > 9-(Methoxycarbonyl)dec-9-enoic acid > 3-(Hepta-1,3-dienyl)hexanedioic acid > 2,3-Dinorthromboxane B1 (Figure 6b). In the R2 period, in the CK_vs._Fm treatment, the down-regulated differential metabolites were arranged in descending order as follows: Methoprotryne > Mitragynine > (-)-Camphoric acid > Coniferyl acetate > Phe(Benzoyl)-Leu-Arg (Figure 6c). In the R2 period, in the CK_vs._Ri treatment, the down-regulated differential metabolites were arranged in the following descending order: Methoprotryne > Mitragynine > 7-Hydroxy-3-(2-hydroxyethyl)-4-methyl-2H-chromen-2-one > (-)-Camphoric acid > 3,3-Dimethylglutaric acid (Figure 6d).

### 2.5. Metabolic Pathway Analysis (KEGG) of Differential Metabolites

The differential metabolites were annotated using the KEGG database. The main differential metabolic pathways triggered by Fm and Ri during the R1 period were the biosynthesis of secondary metabolites and metabolic pathways (Figure 7a,b). During the R2 period, the main differential metabolic pathways triggered by Fm and Ri were microbial metabolism in diverse environments and the degradation of aromatic compounds (Figure 7c,d).

## 3. Discussion

Biomass is a direct indicator of plant growth and productivity. Through mycorrhizal symbionts, plants effectively utilize soil resources, promote photosynthesis and metabolism, and accelerate the accumulation of biomass. In addition, AMF can enhance plant resistance to adversity, including resistance to drought, salinity, and heavy metal toxicity; maintain the normal growth and development of plants; and promote the increase in biomass. This study showed that the biomass of AMF-inoculated licorice increased significantly under saline soil conditions, and the accumulation of plant biomass was different with different inoculants. Liu et al. [23] came to similar conclusions in their research of alfalfa. The results in Wang et al. [24] and Ma et al. [25] showed that the plant biomass increased significantly and the growth state was improved after AMF inoculation.

Roots are the primary parts of plants subjected to salt stress. The root AMF colonization rate represents the symbiosis between plants and mycorrhiza. The area of the roots absorbing nutrients and water is expanded, which promotes the growth and development of plants in stressed soil. The results of this study showed that AMF inoculation affected the root structure of plants in saline soil. The root length, root volume, and number of root branches were all affected to varying degrees, and the root configuration of licorice under different AMF strains was also distinguishing. When plants were inoculated with AMF, the root morphology changed and the root tip necrosis rate was significantly reduced, thus promoting water absorption and root growth [26]. Under AMF treatment, the results of the growth change in *Ammopiptanthus mongolicus* seedlings were consistent in that the treatment promoted biomass accumulation and root growth [27]. Therefore, it is necessary to strengthen the management of AMF in the Hetao area of Inner Mongolia and improve the diversity and colonization rate of AMF.

Broadly targeted metabolomics improves the reliability and accuracy of metabolite identification by comparing the results with tandem mass spectrometry fragmentation patterns and metabolome libraries [28]. In this study, a wide range of targeted metabolomics methods were used to characterize metabolites. The results showed that the trend of metabolic changes in the licorice root inoculated with different AMF strains under saline–alkali stress was inconsistent. High-throughput detection technology was applied to identify the different metabolites inoculated with different AMF strains. In this study, the introduction of Fm at R1 induced 82 differential metabolites (14 up-regulated and 68 down-regulated metabolites), while Ri inoculation induced 73 differential metabolites (20 up-regulated and 53 down-regulated). The introduction of Fm under R2 induced 143 differential metabolites (24 up-regulated metabolites and 119 down-regulated metabolites), while Ri induced 104 differential metabolites (14 up-regulated metabolites and 90 down-regulated metabolites). The results showed that the metabolites of the root exudates of licorice significantly changed under different AMF. The number of differential metabolites increased with time, and the number of up-regulated metabolites and down-regulated metabolites increased significantly. Under saline conditions, 44 up-regulated metabolites and 18 down-regulated metabolites were found in *Puccinellia tenuiflora* plants [29]. By contrast, among the differential metabolites of trifoliate orange mycorrhizal regulation under drought conditions, the number of up-regulated metabolites increased and down-regulated metabolites decreased [30]. The differences in these abiotic results show that AMF altered the secondary metabolism of licorice in response to saline–alkali. Therefore, mycorrhizal changes in plant metabolism depend on the host species as well as the AMF species.

## 4. Materials and Methods

### 4.1. Experimental Design

The experiment was conducted in a greenhouse in Inner Mongolia Agricultural University. Saline soil was taken from the Licorice Experimental Base, Hangjin Banner, Ordos City, Inner Mongolia, China. The preliminary examination of the soil physicochemical properties showed an organic matter (OM) value of 0.73%, pH value of 8.55, and total dissolved salinity (TDS) of 1.86%. In accordance with Cui et al., the TDS of the soil in this experiment was identified as highly saline soil [31]. Therefore, no additional salt stress was added. The test soil was autoclaved (121 °C, 30 min). The selected AMF were *Funneliformis mosseae* (Fm) and *Rhizophagus intraradices* (Ri), which were obtained from the Rhizosphere Biology Research Institute of Yangtze University. Fm and Ri were both amplified by maize in our sterile culture room and were harvested when the colonization rate was over 60%. A mixture of rhizomes and soil with spores were prepared as inoculants.

Three treatment groups were set up in this experiment. They were the control group (CK), the Fm group, and the Ri group, with three replications for each group. The samples were taken at 15 days (R1 period) and 45 days (R2 period) after inoculation and the relevant indexes were determined. 

### 4.2. Plant Culture

The licorice seeds were sterilized and soaked in 1% potassium permanganate solution. After rinsing well in running water, they were treated with soaking and germination. The seeds with cracked shells were raised in substrate (the substrate was autoclaved). Healthy licorice seedlings with the same seedling age were transplanted into 27 × 28 cm pots. Then, 30 g mycorrhizal fungus inoculum was applied around the root of the licorice in each pot. The control group was planted with the above test soil with no inoculation. After the completion of all operations, the licorice seedlings were placed in the laboratory. About 7 days of shade treatment was given, followed by normal management, and sufficient light was provided.

### 4.3. Determinations of Root AMF Colonization Rate, Biomass Production, and Root System Architecture

The licorice plants in different periods were harvested and weighed. The root was cleaned to remove surface impurities. Phenotypic data of related structures were obtained using an EPSON root scanner. The root structure was analyzed by the WinRHIZO System (Regent Instruments INC, Québec City, QC, Canada). The root was cut into 1 cm fragments and Trypan blue staining was used for mycorrhizal staining, as described by Phillips and Hayman [32]. The colonization of AMF in the root was observed under microscope. The root AMF colonization rate was estimated as a percentage of the length of the root segments colonized by AMF in the total observed root segments.

### 4.4. Sample Extraction for Metabolomics Analysis

The root exudates were collected using the solution culture described by Guo et al. [33] and Wang et al. [24]. Selected licorice plants with intact roots were rinsed to remove surface impurities with running water, followed by 1–2 rinses with deionized water. The harvested licorice plants were cultured in a conical bottle for 48 h away from light. The collected solution was the root exudates of the licorice. The filtered liquid was collected in centrifugal tubes. The detection of primary metabolites in the root exudates was conducted using LC-MS/MS metabolomics analysis. The LC-MS/MS analyses were performed using an UHPLC system (Vanquish, Thermo Fisher Scientific, Waltham, MA, USA) with a Phenomenex Kinetex C18 (2.1 mm × 50 mm, 2.6 μm) coupled to an Orbitrap Exploris 120 mass spectrometer (Orbitrap MS, Thermo Fisher Scientific, USA).

A 100 μL sample was taken and mixed with 400 μL of extraction solution (MeOH:ACN, 1:1 (*v*/*v*)). The extraction solution contain deuterated internal standards. The mixed solution was vortexed for 30 s, sonicated for 10 min in a 4 °C water bath, and incubated for 1 h at −40 °C to precipitate proteins. Then, the samples were centrifuged at 12,000 rpm (RCF = 13,800× *g*, R = 8.6 cm) for 15 min at 4 °C. The supernatant was transferred to a fresh glass vial for analysis. The quality control (QC) sample was prepared by mixing an equal aliquot of the supernatant of the samples. The preprocessed data were annotated using the perl program via the HMDB database (V4.0) [34] and we used the database (https://www.kegg.jp/kegg/compound/, accessed on 18 February 2024) [35] to annotate and classify the preprocessed data. Map volcanoes were created using the R (V3.6.2) ggplot2 package. Differential metabolite clustering heatmaps were plotted using the R (V3.6.2) pheatmap package. A PATHWAY FUNCTION ANALYSIS OF DIFFERENTIAL SUBSTANCES WAS PERFORMED USING THE KEGG PATHWAY DATABASE (https://www.kegg.jp/kegg/pathway.html, accessed on 18 February 2024). The Perl program was used to perform pathway enrichment analysis based on the annotation results of all species and the annotation results of differential species, and the enrichment bubble map was plotted [36].

### 4.5. Statistical Analysis

SPSSsoftware (version 26) was used to analyze the variance of biomass, root configuration, and mycorrhizal colonization rate. Duncan’s multiple range test (*p* < 0.05) was used to compare the significant differences between treatments. ProteoWizard (version 3) software was used for metabolite identification and the visual analysis of the original data.

## 5. Conclusions

In conclusion, the effects of different AMF on the root growth and metabolomics of licorice were studied under saline–alkali soil conditions. AMF were found to alleviate licorice growth and increase stress resistance under saline–alkali soil conditions, but different AMF species exerted different effects on the root characteristics and root metabolites. Fm performed better in increasing licorice salinity resistance compared to Ri. Our results provide a new strategy for the application of microbial fertilizers in saline–alkali land. This study not only improves the understanding of the ecological adaptability of licorice, but also provides new ideas for saline–alkali land agriculture and medicinal plant cultivation. 

## Figures and Tables

**Figure 1 plants-13-02652-f001:**
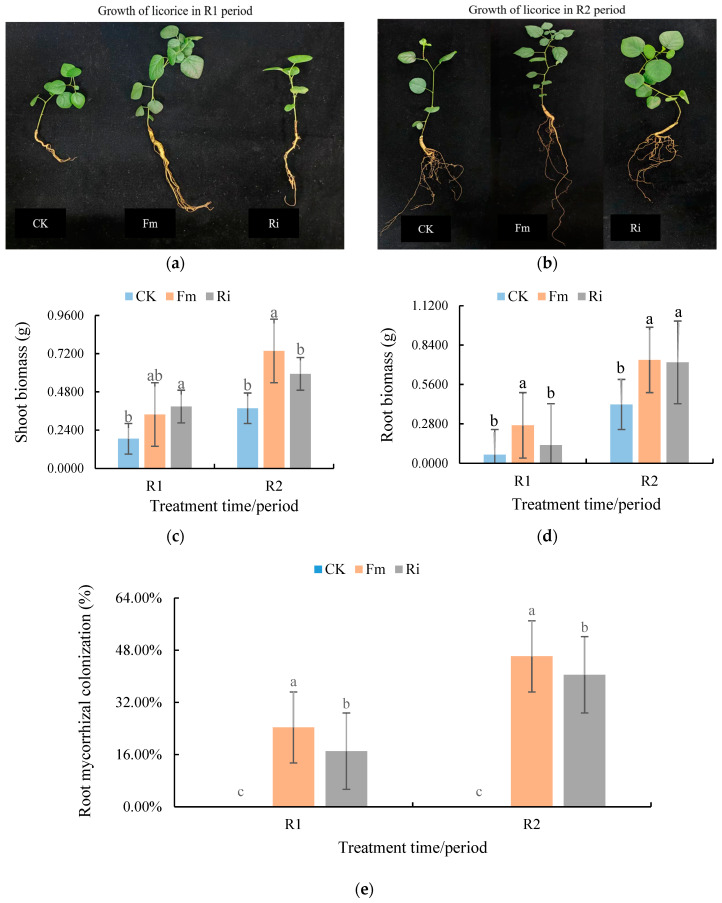
Changes in plant growth performance for R1 period (**a**) and R2 period (**b**); shoot (**c**) and root (**d**) biomass production; and root mycorrhizal colonization rate (**e**). Data followed by different letters above the bars indicate significant (*p* < 0.05) differences. Abbreviations: R1 and R2, the different periods; CK, treatment with no inoculation; Fm, treatment with *Funneliformis mosseae* inoculation; Ri, treatment with *Rhizophagus intraradices* inoculation.

**Figure 2 plants-13-02652-f002:**
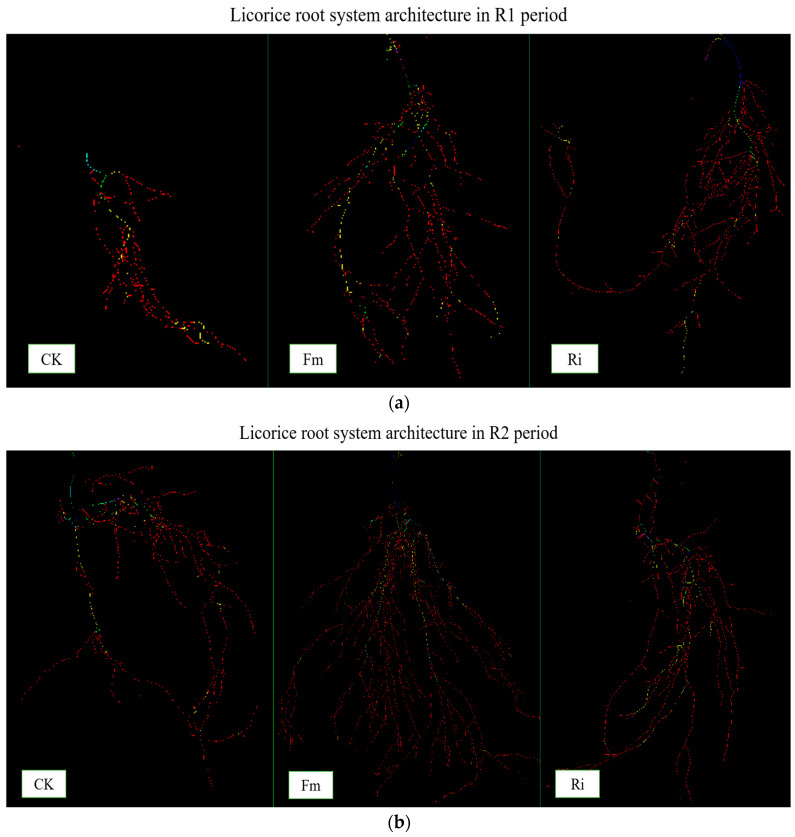
Changes in root system architecture in R1 period (**a**) and R2 period (**b**). Abbreviations: R1 and R2, the different periods; CK, treatment with no inoculation; Fm, treatment with *Funneliformis mosseae* inoculation; Ri, treatment with *Rhizophagus intraradices* inoculation.

**Figure 3 plants-13-02652-f003:**
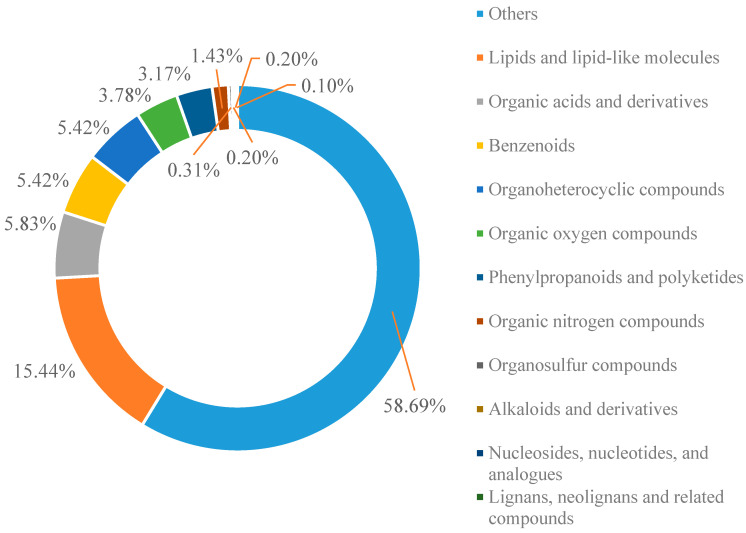
Classification statistics of compounds in the root of licorice.

**Figure 4 plants-13-02652-f004:**
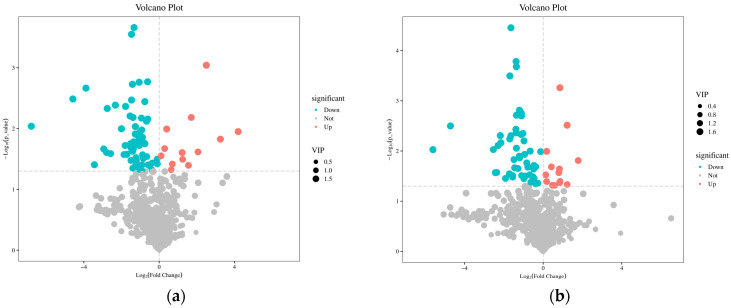
Number of differential metabolites in roots of licorice in R1 period for CK_vs._Fm (**a**) and CK_vs._Ri (**b**) treatments, and in R2 period for CK_vs._Fm (**c**) and CK_vs._Ri (**d**) treatments. Abbreviations: R1 and R2, the different periods; CK, treatment with no inoculation; Fm, treatment with *Funneliformis mosseae* inoculation; Ri, treatment with *Rhizophagus intraradices* inoculation.

**Figure 5 plants-13-02652-f005:**
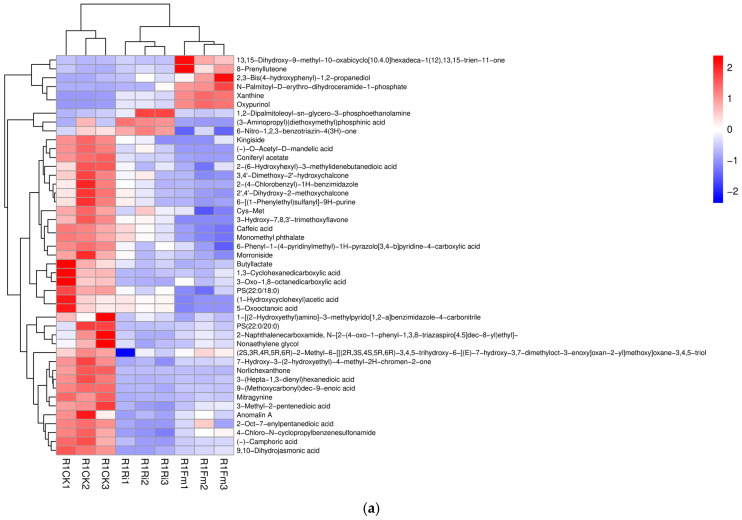
Heatmap of metabolites in licorice root in R1 period for CK_vs._Fm_vs._Ri (**a**) treatment, and in R2 period for CK_vs._Fm_vs._Ri (**b**) treatment. Abbreviations: R1 and R2, the different periods; CK, treatment with no inoculation; Fm, treatment with *Funneliformis mosseae* inoculation; Ri, treatment with *Rhizophagus intraradices* inoculation.

**Figure 6 plants-13-02652-f006:**
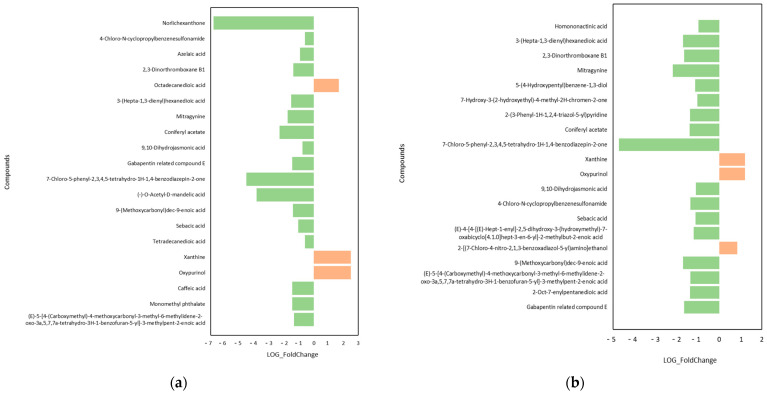
Top 20 differential metabolites in roots of licorice in R1 period for CK_vs._Fm (**a**) and CK_vs._Ri (**b**) treatments, and in R2 period for CK_vs._Fm (**c**) and CK_vs._Ri (**d**) treatments. Abbreviations: R1 and R2, the different periods; CK, treatment with no inoculation; Fm, treatment with *Funneliformis mosseae* inoculation; Ri, treatment with *Rhizophagus intraradices* inoculation.

**Figure 7 plants-13-02652-f007:**
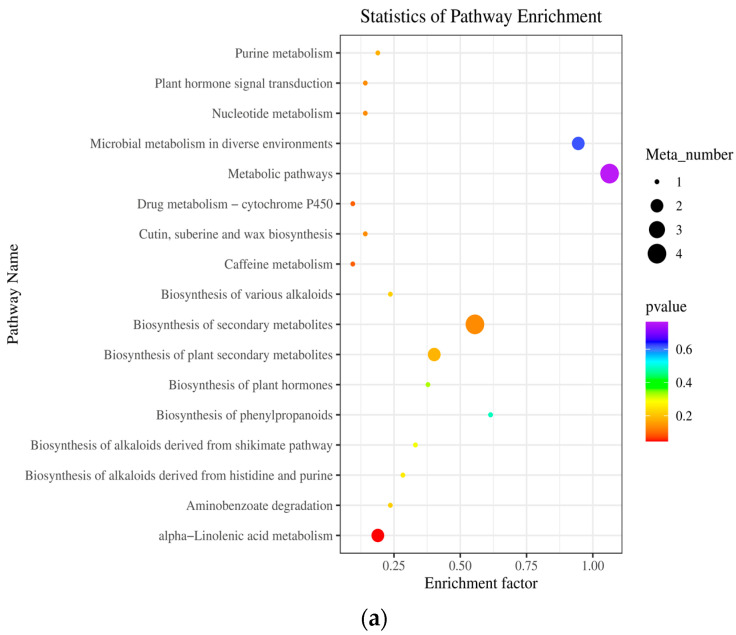
Top 20 metabolic pathways of differential metabolites annotated by KEEG in roots of licorice between R1 period for CK_vs._Fm (**a**) and CK_vs._Ri (**b**) treatments and R2 period for CK_vs._Fm (**c**) and CK_vs._Ri (**d**) treatments. Abbreviations: R1 and R2, the different periods; CK, treatment with no inoculation; Fm, treatment with *Funneliformis mosseae* inoculation; Ri, treatment with *Rhizophagus intraradices* inoculation.

**Table 1 plants-13-02652-t001:** Morphological traits of root systems of self-rooted licorice plants. Abbreviations: R1 and R2, the different periods; CK, treatment with no inoculation; Fm, treatment with *Funneliformis mosseae* inoculation; Ri, treatment with *Rhizophagus intraradices* inoculation.

Parameters	Treatments	Treatment Time/Period	
		R1	R2
Root total length (cm)	CK	24.52 ± 2.39 c	86.35 ± 2.92 c
	Fm	47.35 ± 5.61 a	177.46 ± 18.51 a
	Ri	35.93 ± 1.86 b	120.85 ± 16.30 b
Root average diameter (mm)	CK	0.37 ± 0.04 b	0.57 ± 0.07 b
	Fm	0.50 ± 0.08 a	0.84 ± 0.05 a
	Ri	0.46 ± 0.05 ab	0.72 ± 0.07 a
Root total surface area (cm^2^)	CK	3.84 ± 0.22 c	14.85 ± 2.00 b
	Fm	13.01 ± 2.28 a	28.61 ± 4.86 a
	Ri	7.10 ± 1.33 b	16.82 ± 1.05 b
Root volume (cm^3^)	CK	0.06 ± 0.01 c	0.25 ± 0.03 c
	Fm	0.21 ± 0.02 a	0.54 ± 0.08 a
	Ri	0.14 ± 0.03 b	0.38 ± 0.03 b
Number of root branches (n.)	CK	56.00 ± 14.53 b	218.00 ± 35.00 c
	Fm	98.67 ± 21.55 a	495.33 ± 30.73 a
	Ri	81.33 ± 5.13 ab	319.00 ± 31.00 b

Different letters in the table indicate significant differences at the 5% (*p* < 0.05) level.

## Data Availability

The data are contained within the manuscript.

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
