# Peer review of "Metabolomics Revealed the Tolerance and Growth Dynamics of Arbuscular Mycorrhizal Fungi (AMF) to Soil Salinity in Licorice"

_plants, 2024, doi:10.3390/plants13182652_

Round 1

Reviewer 1 Report

Comments and Suggestions for Authors

In the m/s “Metabolomics Revealed the Tolerance and Growth Dynamics 2 of Arbuscular Mycorrhizal Fungi (AMF) to Soil Salinity in Licorice”, the authors highlighted the metabolic analysis of licorice salinity tolerance through inoculation with AMF, Funneliformis mosseae (Fm); and Rhizophagus intraradices (Ri). Numerous studies have pointed to the role of AMF in improving plant tolerance to salt stress. Please indicate the originality of your work. In addition, many sentences need to be revised linguistically. They are difficult to understand.

The abstract is not well structured, the objectives are not clear. The methodology was not well presented, experimental procedures, tools and techniques used. Please use simple and direct language. Abstract provides a condensed overview of the article, allowing readers to quickly understand the key points without reading the whole document.

 - L18: abiotic stress... specify the stress studied

-  L20: Please give the complete name of Fm and Ri, as are the first time of rinsing it in the m/s.

Introduction

If possible, add a brief overview of factors that support increased saline level in agricultural lands, and some few effects (symptoms) of salt stress in Licorice.

it is important to add some publications of specific work on Funneliformis mosseae, the and Rhizophagus intraradices., which has been used either to alleviate salt stress in plants, or on licorice. If this is the first work in this area, it should be mentioned.

L74-79. I think your objective is to analyse the metabolites of licorice under salt and alkaline stress, as well as in the presence of the effect of AMF. You should therefore add AMF.

M&M

- L263. Can the authors explain the meaning of this sentence “more than 60% of each fungus was infested before harvest”?

- Can the authors explain how they applied salt stress? Please indicate the characteristics of the soil used, particularly the salt content, and the level of salt stress that the licorice can tolerate. Did you apply severe or moderate stress?

Results and discussion:

When the authors quantified AMF colonisation, they did not seem to consider arbuscules, which are the functional structure of symbioses. Please add data on the % of arbuscules in the different treatments?

Conclusion

L309. drought stresses or salt stress ?

It is recommended that you reformulate your conclusions, as your current version does not highlight the novelty of your work. Summarises your main discoveries, their significance, especially in sustainable agriculture and future prospects.

References

Add the missing DOI and check all references citation in the text and list (volume number and pages).

Author Response

Response to Reviewer 1 Comments

1. Summary

Thank you very much for taking time to review this manuscript. Please find the detailed responses below and the corresponding revisions/corrections highlighted/in track changes in the re-submitted files.

2. Questions for General Evaluation

Reviewer’s Evaluation

Response and Revisions

Does the introduction provide sufficient background and include all relevant references?

Can be improved

Is the research design appropriate?

Can be improved

Are the methods adequately described?

Can be improved

Are the results clearly presented?

Can be improved

Are the conclusions supported by the results?

Can be improved

3. Point-by-point response to Comments and Suggestions for Authors

Comments 1: In the m/s “Metabolomics Revealed the Tolerance and Growth Dynamics 2 of Arbuscular Mycorrhizal Fungi (AMF) to Soil Salinity in Licorice”, the authors highlighted the metabolic analysis of licorice salinity tolerance through inoculation with AMF, Funneliformis mosseae (Fm); and Rhizophagus intraradices (Ri). Numerous studies have pointed to the role of AMF in improving plant tolerance to salt stress. Please indicate the originality of your work. In addition, many sentences need to be revised linguistically. They are difficult to understand.

The abstract is not well structured, the objectives are not clear. The methodology was not well presented, experimental procedures, tools and techniques used. Please use simple and direct language. Abstract provides a condensed overview of the article, allowing readers to quickly understand the key points without reading the whole document.

 - L18: abiotic stress... specify the stress studied

-  L20: Please give the complete name of Fm and Ri, as are the first time of rinsing it in the m/s.

Response 1: Thank you for pointing this out. We agree with this comment. Therefore, we have enriched the whole introduction part,detailed the experimental methodology.

And regarding the question you mentioned, we ‘ve changed the abiotic stress to soil salinity ---Line 21.

And added the full names of Fm and Ri when they first appeared. ---Line 19-20

Comments 2: If possible, add a brief overview of factors that support increased saline level in agricultural lands, and some few effects (symptoms) of salt stress in Licorice.

it is important to add some publications of specific work on Funneliformis mosseae, the and Rhizophagus intraradices., which has been used either to alleviate salt stress in plants, or on licorice. If this is the first work in this area, it should be mentioned.

L74-79. I think your objective is to analyse the metabolites of licorice under salt and alkaline stress, as well as in the presence of the effect of AMF. You should therefore add AMF.

Response 2: Agree. We have, accordingly modified to emphasize this point. And a section has been added to explain it. ---Line79-85

Comments 3: - L263. Can the authors explain the meaning of this sentence “more than 60% of each fungus was infested before harvest”?

- Can the authors explain how they applied salt stress? Please indicate the characteristics of the soil used, particularly the salt content, and the level of salt stress that the licorice can tolerate. Did you apply severe or moderate stress?

Response 3: Agree. We have, accordingly modified.to emphasize this point. We're sorry for the misunderstanding caused by our carelessness, also we were surprised by this statement, so we've removed this content to ensure the integrity of the article

Comments 4: When the authors quantified AMF colonisation, they did not seem to consider arbuscules, which are the functional structure of symbioses. Please add data on the % of arbuscules in the different treatments?

Response 4: Thank you for pointing this out. We scanned the entire root after inoculation with AMF, and the result was a complete result, which included bush branches. At the same time, the changes under different treatments are reflected in Table 1 of the paper.

Comments 5: L309. drought stresses or salt stress ?

It is recommended that you reformulate your conclusions, as your current version does not highlight the novelty of your work. Summarises your main discoveries, their significance, especially in sustainable agriculture and future prospects.

Response 5: Thank you for pointing this out. We agree with this comment. And this content has been corrected in saline-alkaline stresses. ---Line324

And this article does not additionally address salt stress, which refers to the degree of salinity of the soil itself, and has been re-added to clarify this point. ---Line270-271, Line292-294

Comments 6: Add the missing DOI and check all references citation in the text and list (volume number and pages).

Response 6: Agree. We have, accordingly modified.to emphasize this point. We have added some of the missing DOI, but we have been able to find some DOI in some articles.

Reviewer 2 Report

Comments and Suggestions for Authors

Your method section is the weakest part of your paper. You did not give certain details that should be mentioned. For example: the enrichment plots are generated by metaboanalyst. You did not cite this. It is obvious that MetaboAnalyst was used for Figures 4,5 and 7. You should always mention everything you did. I am aware that you are going for brevity, but it is best to explain what you did or give in details what you did in the method. Even if you cite from another paper.

For metabolomics datasets, you always need some supplemental information.

You also should indicate the level of identification of metabolites using the metabolomics standards initiative (MSI). You may not be able to show all, but at least the most important ones, especially if you got fragmentation data or not, and the level of identification.

You also generated plots, but you never mentioned what tool was used in the materials and method section. It seemed to be generated by MetaboAnalyst. Please cite this resource as well.

You also grouped the metabolites into different classes, but you never said how you did so. I recognize this as using ClassyFire to classify your metabolites into groupings. Please cite this, as you never mentioned it.

Also consider changing some things in the materials and method section. By that I mean, some areas are not written in past tense.

Also, a word of advice: if you are saying compounds are up-or-downregulated, these are features, not metabolites. When you confirmed them, then they are putative or tentative metabolites. Only when you have standards can you really say with certainty that they are metabolites, especially when fragmented the same way.

Using LC-MS/MS, you should always be prepared to give tables of the fragments of putative metabolites that you generated. That way, others can check if they get the same fragments as you if they want to use your methodology.

Also, you mentioned using isotopic labeled standards in your samples, but you never mentioned what they are.

figure 1: you should mention statistical significance for this figure 1 a and b.

figure 3: since classyfire was used for this, mention this in the title of figure 3.

figure 6: some names of the compounds here are cut off. please fix this, or find a shorter alternative name for this using Pubchem or another resource. Remember to cite every tool used, especially KEGG.

Table 1: you mentioned a, b and c as significant values, but you never provided what a, b, or c are in terms of p-value. I understand that a table is most appropriate for this but make a note of the significant values of a, b or c.

What instrument did you use for LC-MS/MS? None of this was provided, or any acquisition parameters. You should include this as well.

For the introduction, did you have any hypotheses? How unique is your approach? Emphasizing these would really make the introduction stand out more.

With these changes, the manuscript will be greatly improved.

Author Response

Response to Reviewer 2 Comments

1. Summary

Thank you very much for taking time to review this manuscript. Please find the detailed responses below and the corresponding revisions/corrections highlighted/in track changes in the re-submitted files.

2. Questions for General Evaluation

Reviewer’s Evaluation

Response and Revisions

Does the introduction provide sufficient background and include all relevant references?

Yes

Is the research design appropriate?

Yes

Are the methods adequately described?

Must be improved

Are the results clearly presented?

Can be improved

Are the conclusions supported by the results?

Yes

3. Point-by-point response to Comments and Suggestions for Authors

Comments 1: Your method section is the weakest part of your paper. You did not give certain details that should be mentioned. For example: the enrichment plots are generated by metaboanalyst. You did not cite this. It is obvious that MetaboAnalyst was used for Figures 4,5 and 7. You should always mention everything you did. I am aware that you are going for brevity, but it is best to explain what you did or give in details what you did in the method. Even if you cite from another paper.

For metabolomics datasets, you always need some supplemental information.

Response 1: Thank you for pointing this out. We agree with this comment. And the corresponding drawing tools have been added, and refine some of the methods.

Comments 2: You also should indicate the level of identification of metabolites using the metabolomics standards initiative (MSI). You may not be able to show all, but at least the most important ones, especially if you got fragmentation data or not, and the level of identification.

You also generated plots, but you never mentioned what tool was used in the materials and method section. It seemed to be generated by MetaboAnalyst. Please cite this resource as well.

Response 2: Agree. We have, accordingly modified.to emphasize this point. The specific tools and methods used in the drawing have been added to the article. ---Line 316-323

Comments 3: You also grouped the metabolites into different classes, but you never said how you did so. I recognize this as using ClassyFire to classify your metabolites into groupings. Please cite this, as you never mentioned it.

Also consider changing some things in the materials and method section. By that I mean, some areas are not written in past tense.

Also, a word of advice: if you are saying compounds are up-or-downregulated, these are features, not metabolites. When you confirmed them, then they are putative or tentative metabolites. Only when you have standards can you really say with certainty that they are metabolites, especially when fragmented the same way.

Response 3: Agree. We have, accordingly modified.to emphasize this point. Regarding the question of metabolite grouping that you mentioned, we have already added to it. ---Line318-320

And regarding the tense of the article you mentioned, we had also re-improved the content of materials and methods. ---Line 321-326

Comments 4: Using LC-MS/MS, you should always be prepared to give tables of the fragments of putative metabolites that you generated. That way, others can check if they get the same fragments as you if they want to use your methodology.

Also, you mentioned using isotopic labeled standards in your samples, but you never mentioned what they are.

Response 4: Thank you for pointing this out. With regard to your question about the isotopes of starvation, we believe that this is only a quantitative enhancement and is not the focus of this experiment

Comments 5: figure 1: you should mention statistical significance for this figure 1 a and b.

figure 3: since classyfire was used for this, mention this in the title of figure 3.

figure 6: some names of the compounds here are cut off. please fix this, or find a shorter alternative name for this using Pubchem or another resource. Remember to cite every tool used, especially KEGG.

Table 1: you mentioned a, b and c as significant values, but you never provided what a, b, or c are in terms of p-value. I understand that a table is most appropriate for this but make a note of the significant values of a, b or c.

Response 5: Thank you for pointing this out. We agree with this comment. Regarding some of the issues you mentioned about the prominent starvation of charts, we have corrected and supplemented them below the corresponding icons. ----Line 96, 124

And the classification question you mentioned, it is to pave the way for the subsequent research work on functions, etc.

As for the truncated names of some substances in Figure 6 that you mentioned, I'm sorry that this was a careless part of our work, and now we have refilled the missing content in the figure to ensure the completeness of the figure.

Comments 6: What instrument did you use for LC-MS/MS? None of this was provided, or any acquisition parameters. You should include this as well.

For the introduction, did you have any hypotheses? How unique is your approach? Emphasizing these would really make the introduction stand out more.

Response 6: Agree. We have, accordingly modified.to emphasize this point. Regarding the instrument issue you mentioned, we have made additions and changes in the appropriate places.

We've made detailed changes to the content you mentioned. ----Line 77-87

Round 2

Reviewer 1 Report

Comments and Suggestions for Authors

the authors' responses are not convincing.  

many comments are not answered ( Ex. Please indicate the originality of your work. In addition, many sentences need linguistic revision. They are difficult to understand. no answer is found for this question. the same for many other comments...).  

I suggested that specialists assess the English quality of this document.

Round 3

Reviewer 1 Report

Comments and Suggestions for Authors

Dear authors,

Thank you for your efforts and thoughtful revisions of the manuscript. The additional data you provided in response to my comments strengthened the manuscript. I believe the manuscript is now ready for publication. I recommend its acceptance after minor clarifications in the abstract, adding the meaning of the R1 and R2 periods, in order to provide a condensed overview of the article, allowing readers to quickly understand the key points without having to read the entire article. 

Thank you once again for your efforts in revising the manuscript.

Yours sincerely

Author Response

Comments 1: adding the meaning of the R1 and R2 periods, in order to provide a condensed overview of the article, allowing readers to quickly understand the key points without having to read the entire article.

Response 1: Thank you for your valuable feedback. We have added the meaning of the R1 and R2 periods in abstract in the manuscript. See line 17-18.